# Categorical identity signatures can reduce host error rates during brood parasitism

Tanmay Dixit[1,2]*, Ming Liu[3], Jana M. Riederer[4], Jonah M. Walker[1,5], Cameron J. Blair[2], Jess Lund[1,2], Collins Moya[6], Claire N. Spottiswoode[1,2]

1 Department of Zoology, University of Cambridge, Cambridge, United Kingdom, 2 FitzPatrick Institute of African Ornithology, Department of Biological Sciences, University of Cape Town, Rondebosch, Cape Town, South Africa, 3 Department of Biology, University of Oxford, Oxford, United Kingdom, 4 Groningen Institute for Evolutionary Life Sciences, University of Groningen, Groningen, The Netherlands, 5 Tree of Life Programme, Wellcome Sanger Institute, Hinxton, United Kingdom, 6 Musumanene Farm, Choma, Zambia

* td349@cam.ac.uk

## Abstract

Biological recognition is often modeled as involving discrimination of continuously-distributed (and continuously-perceived) traits according to decision thresholds. However, traits such as animal signals can be categorically distributed. Here, we test how such categorical distributions may influence fundamental trade-offs in signal recognition, using a brood parasite–host system involving identity recognition. The African cuckoo finch *Anomalospiza imberbis* parasitizes several host species, each of which has evolved inter-individual variation in egg appearance ("egg signatures") that facilitates recognition and rejection of mimetic cuckoo finch eggs. We demonstrate that egg signature traits in one host species, the zitting cisticola *Cisticola juncidis*, are categorically distributed. Field experiments reveal that zitting cisticolas make fewer Type II errors (accepting parasitic eggs) *and* Type I errors (rejecting their own eggs) than hosts exhibiting continuous variation. This challenges the long-standing expectation (from classification models, statistics, and signal detection theory) of a strict trade-off between these two error types. Individual-based simulations clarify mechanisms by which categorical variation can generate low error rates, especially when combined with "category-based rejection", whereby hosts only reject eggs of different categories to their own. Our findings show that the categorical distribution and category-based perception of trait variation can shape error trade-offs and coevolutionary dynamics, which should inform studies on other mimicry or self/non-self recognition systems, including immune recognition. They also highlight the importance of quantifying trait distributions and how they are perceived, when understanding coevolution between deceivers and those they deceive.

**Data availability statement:** All data and code are available at https://doi.org/10.17863/CAM.116928.2.

**Funding:** TD was funded by a Rosemary Grant Advanced Award from the Society for the Study of Evolution (https://www.evolutionsociety.org/); a Balfour Studentship from the Department of Zoology, University of Cambridge; and St John's College, Jesus College, and the School of Biological Sciences, University of Cambridge. JMR was funded by a University of Groningen Adaptive Life Scholarship and additionally received funding through a European Research Council (ERC) advanced grant (Grant no. 789240 with recipient FJ Weissing; https://erc.europa.eu/homepage). CNS was funded by a BBSRC David Phillips Fellowship (Grant Number BB/J014109/1; https://www.ukri.org/councils/bbsrc/) and a Royal Society Dorothy Hodgkin Fellowship (https://royalsociety.org/). None of these sponsors or funders played any role in the study design, data collection and analysis, decision to publish, or preparation of the manuscript.

**Competing interests:** The authors have declared that no competing interests exist.

**Abbreviations:** MHC, Major Histocompatibility Complex; PAM, Partitioning Around Medioids; SDT, signal detection theory; SEs, standard errors.

## Introduction

Variation in nature is generally continuous, with many biological traits such as human height distributed approximately as normal distributions without categories [1]. Similarly, in sensory ecology, models generally assume that signal variation is continuous and normally distributed, and is perceived in a continuous manner [2–10].

Nevertheless, neither variation nor its perception need be continuous or normally distributed. "Categorically-distributed" traits comprise discrete polymorphisms: individuals exhibit one of two or more discrete traits (e.g., color polymorphisms where each individual belongs to a particular color "morph"), with no or few intermediates [11]. Such traits are "multimodal", meaning that the distribution of trait values includes more than one peak. Similarly, trait perception need not be continuous; for instance, categorical perception (where animals respond to stimuli as though they belong in discrete categories [4]) may be involved in mate choice [12], perception of vocalisations [13], and egg discrimination by hosts of brood parasites [14]. While modeling has suggested that signals of quality should exhibit unimodal, continuous distributions, and signals of identity may exhibit complex, multimodal distributions [15], eco-evolutionary consequences of these distributions remain unknown. Here, we use empirical and simulation approaches to study categorical versus continuous variation in a context where signals of identity have evolved: brood parasitism.

Avian brood parasites lay eggs in other birds' (hosts') nests, offloading parental care onto hosts. Costs of being parasitized can select for egg discrimination and rejection by hosts and egg mimicry by parasites [16]. In response to mimicry, some hosts have evolved inter-individual variation in egg color and pattern. These "signatures" of identity mean that individual parasitic eggs cannot mimic all host eggs [17,18]. Some host signature traits are continuously-distributed, such as the number of markings on eggs in tawny-flanked prinias *Prinia subflava* [19]. Other hosts exhibit categorical variation: among hosts of common cuckoos *Cuculus canorus*, Daurian redstart *Phoenicurus auroreus* eggs are either blue or pink [20]; ashy-throated parrotbill *Paradoxornis alphonsianus* eggs are blue or white [21].

Different trait distributions may have important coevolutionary consequences if they affect the frequencies of two types of error that hosts make. By accepting parasitic eggs (analogous to Type II errors in statistics), hosts suffer costs of parasitism [16,17]. Hosts sometimes also reject their own eggs (Type I errors), suffering reduced clutch sizes [22,23]. Rates of these errors should depend on two properties of hosts. First, signal detection theory (SDT; a framework describing how perceivers discriminate signals from other signals or background noise) proposes that error rates depend on how sensitive perceivers are to small differences [7,9]. Specifically, hosts capable of discriminating small differences between eggs (e.g., those with stricter recognition thresholds) should make fewer Type II errors and more Type I errors than hosts with poorer recognition abilities [23,24]. In other words, sensitivity to small differences between eggs mediates a trade-off between Type I and Type II errors, just as sensitivity mediates Type I:Type II error rate trade-offs in classification models and statistics [7,9,16,23]. Second, error rates depend upon host variation: if females lay consistent egg signatures (i.e., intra-individual variation is low), this will reduce Type I

error frequency. If each female's signature is distinct from other females' signatures (i.e., inter-individual variation is high), this will reduce Type II error frequency [22]. However, empirical observations have shown that consistency and distinctiveness in egg appearance are negatively correlated, likely for mechanistic reasons [22,25] (also see S1 Text §e), which additionally generates a Type I:Type II error rate trade-off.

Given these trade-offs, can frequencies of both types of error be reduced simultaneously? Here, we suggest that distribution shapes (categorical versus continuous) can affect error rates. One potential mechanism, generalizable to other systems but described here in the context of egg rejection, is the following. In species exhibiting categorical variation in signature traits, a given host female's eggs will likely belong to a single category. Thus, if such hosts generally reject only eggs of different categories, they should make few Type I errors (i.e., rarely reject their own eggs). By contrast, for species exhibiting continuous variation, no categories exist; thus, decision rules such as "accept only eggs of the same category" cannot apply. This may increase Type I error rates, because aberrant eggs within their clutches may be misidentified as parasitic eggs. Moreover, trait distribution shapes will influence differences between parasitic eggs and host eggs, and thus the frequency of Type II errors (accepting parasitic eggs), although it is difficult to intuit which distribution shapes should be associated with reduced Type II error rates.

In southern Zambia, different gentes (maternal lineages) of the cuckoo finch *Anomalospiza imberbis* parasitize different species of cisticolid warbler, each of which lays eggs with female-specific signatures [22,26]. Zitting cisticola (*Cisticola juncidis*) egg traits appear to be categorically-distributed (Fig 1a). Croaking cisticolas (*C. natalensis*) lay eggs that appear to comprise a mix of categorical and continuous traits. Tawny-flanked prinias (Fig 1b) and red-faced cisticolas (*C. erythrops*) lay eggs that appear to exhibit continuous variation across color and pattern traits.

Inspired by this diversity, here we combine empirical and modeling approaches to quantify host trait distributions and test the hypothesis that trait distributions affect error rates, with important consequences for hosts' success in coevolutionary arms races with parasites.

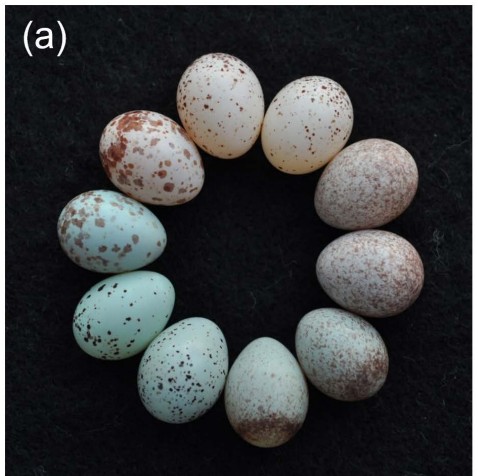 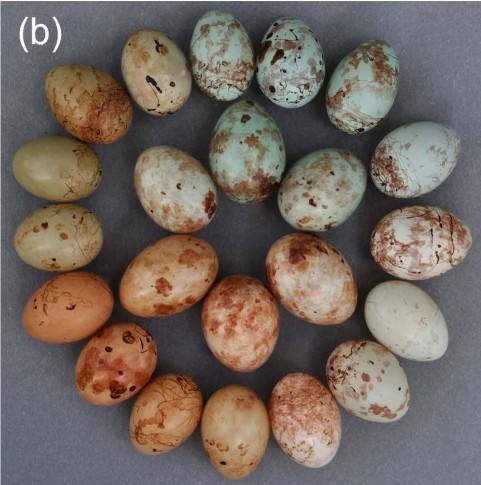

**Fig 1. Eggs of the two most common hosts of the cuckoo finch at our study site. (a)** Zitting cisticola eggs, which to the human eye vary categorically in background color (hereafter, color: blue or white), marking size (small speckles or large blotches), and marking luminance (pale or dark). Since no eggs have dark blotches, there are six categories to the human eye. These are, clockwise from top: white with dark speckles; white with pale speckles; blue with pale speckles; blue with dark speckles; blue with pale blotches; white with pale blotches. Two eggs of each trait combination are shown, except for blue with pale blotches and white with pale blotches, for which only one egg of each is shown. Cuckoo finch eggs laid in zitting cisticola nests mimic zitting cisticola phenotypes (Fig A in S1 Text). **(b)** Tawny-flanked prinia eggs (outer circle), which, to the human eye show continuous variation, and eggs laid in prinia nests by the maternal lineage of cuckoo finches which specializes on prinias (inner circle). Images of eggs of the other species that are parasitized at the study site, croaking cisticolas and red-faced cisticolas, are shown in Fig 2. Images: TD and CNS.

## Results

In field experiments, we replaced a host egg from each host nest with a conspecific "experimental" egg from a different nest to simulate parasitism by cuckoo finches. For each host species, these experiments allowed us to (a) identify the egg traits for which a difference between the host egg trait and the experimental egg trait predicted rejection, to quantify the distributions of these traits in hosts, and to quantify rates of (b) Type II errors (accepting parasitic eggs), and (c) Type I errors (rejecting own eggs). We then (d) used simulations to investigate how categorical variation influences error rates, finding two mechanisms by which categorical variation can reduce error rates.

### a. Distributions of traits associated with rejection in each host species

**i. Zitting cisticolas.** Differences between host and experimental eggs in three traits predicted rejection in zitting cisticolas: color (Estimate = 0.80 ± 0.22SE, $Z_{86}$ = 3.60, $P$ < 0.001), marking luminance (Estimate = 0.60 ± 0.19SE, $Z_{86}$ = 3.16, $P$ = 0.002), and mean marking size (Estimate = 2.22 ± 0.85SE, $Z_{86}$ = 2.60, $P$ = 0.009; Nagelkerke's $R^2$ = 0.70; Table 1). To the human eye, each of these three traits is categorically distributed in this species, with six observed trait combinations (Figs 1a and 2a). Almost all (51/52) foreign eggs that differed from the host clutch in one or more human-assigned category were rejected; almost all (35/38) eggs that did not differ in any category were accepted. Partitioning Around Medioids (PAM) cluster analyses ($n$ = 119 for color, $n$ = 400 for pattern traits) showed that indeed each trait was best described as two clusters, with the number of clusters differing significantly from one (all $P$ < 0.01; Table 1), and with each cluster corresponding well to the human categorization of phenotypes (Fig 2a). These results were supported by direct tests for multimodality (S1 Text §d). In summary, zitting cisticolas showed categorical variation in traits predicting rejection.

**ii. Croaking cisticolas.** Differences between host and experimental eggs in three traits predicted rejection in croaking cisticolas: color (Estimate = 0.79 ± 0.25SE, $Z_{47}$ = 3.18, $P$ = 0.002), pattern coverage (Estimate = 30.83 ± 14.46SE, $Z_{47}$ = 2.13, $P$ = 0.03; Nagelkerke's $R^2$ = 0.43 for this model), and marking luminance (which, alongside color, predicted rejection when eggs were not immaculate: Estimate = 0.38 ± 0.16SE, $Z_{35}$ = 2.38, $P$ = 0.02; Nagelkerke's $R^2$ = 0.76 for this model excluding immaculate eggs; see S1 Text §b for details; also see Table 1). A PAM cluster analysis ($n$ = 166 for marking luminance; n = 194 for pattern coverage) showed that for both traits the number of clusters did not differ significantly

**Table 1. Summary of results from egg rejection experiments and cluster analyses.**

| Species | Trait (sample size) | Estimate ±SE (*P*-value) from rejection model | Best-supported number of clusters (*P*-value from Duda–Hart test) |
|---|---|---|---|
| Zitting cisticola | Color (119) | 0.80 ± 0.22 (<0.001) | 2 (0.002) |
| | Marking luminance (400) | 0.60 ± 0.19 (0.002) | 2 (<0.001) |
| | Marking size (400) | 2.22 ± 0.85 (0.009) | 2 (<0.001) |
| Croaking cisticola | Color (73) | 0.79 ± 0.25 (0.002) | 2 (0.03) |
| | Pattern coverage (194) | 30.83 ± 14.46 (0.03) | 1 (0.51) |
| | Marking luminance (166) | 0.38 ± 0.16 (0.02) | 1 (0.16) |
| Tawny-flanked prinia | Color (254) | 0.27 ± 0.06 (<0.001) | 1 (0.06) |
| | Pattern coverage (511) | 15.46 ± 4.56 (<0.001) | 1 (0.18) |
| | Pattern dispersion (511) | 3.38 ± 2.12 (0.11) | 1 (0.89) |
| Red-faced cisticola | Marking luminance (182) | 0.30 ± 0.12 (0.02) | 1 (0.44) |
| | No. SIFT features (182) | 0.03 ± 0.01 (0.01) | 1 (0.09) |

For each species, model outputs (estimate, standard error, *P*-value) for each trait in the final rejection model are provided, as are results of the cluster analysis (best-supported number of clusters, and the output of the Duda–Hart test, which tests whether the number of clusters differs significantly from one; see Methods). In rejection models, predictor variables were absolute differences in trait values between each host clutch and the conspecific "experimental" egg placed inside that host nest (see Methods), whereas cluster analyses were conducted on host trait values rather than trait differences.

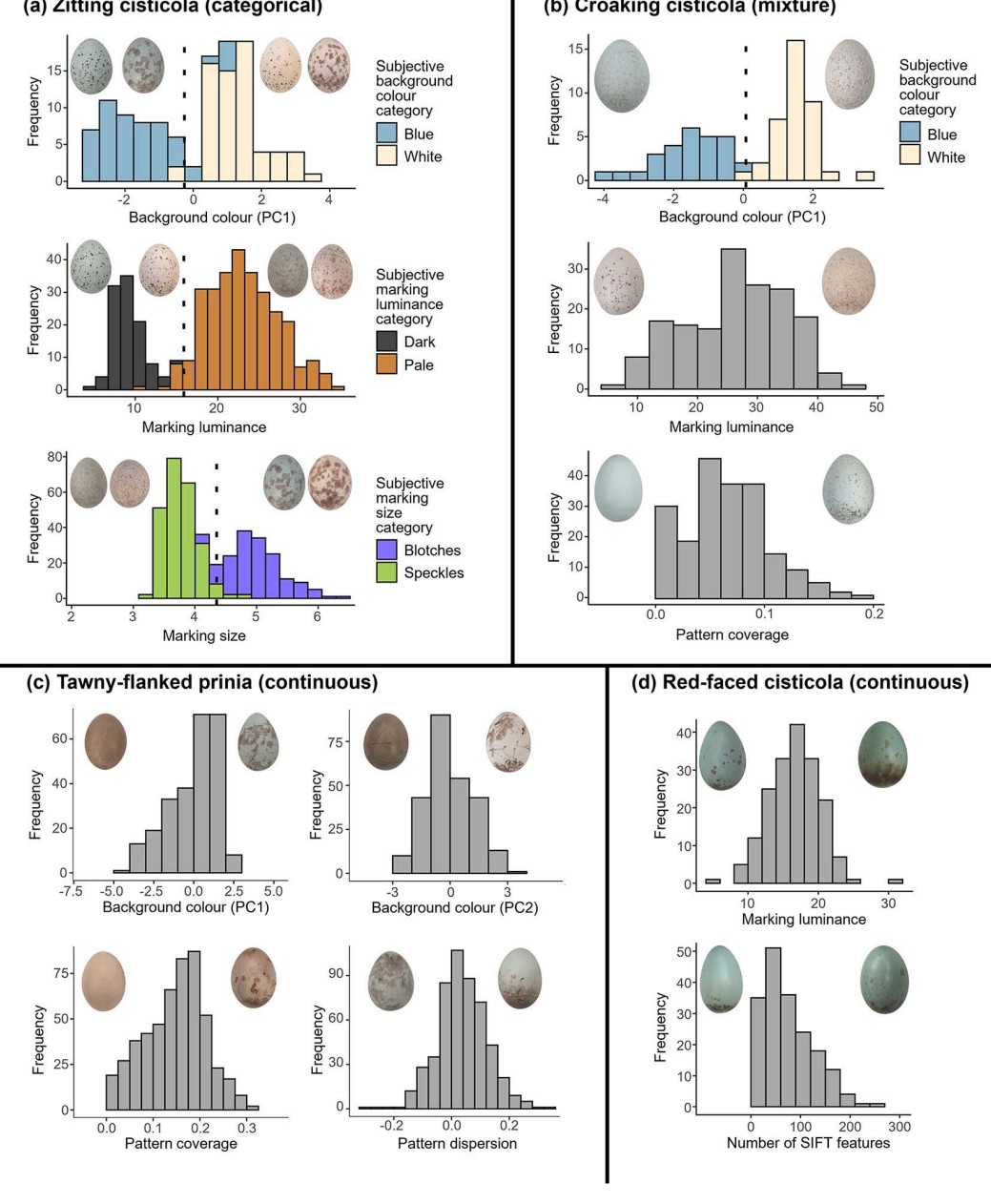

**Fig 2. Distributions of the traits predicting rejection in the four hosts studied.** Histograms illustrating categorical variation (all traits in **(a)**; one trait in **(b)**) include dotted lines indicating the cutoffs between the two clusters, and bars colored according to the proportion of each bar corresponding to visual categorization of each trait. These illustrate that the clusters corresponded well to human categorization of phenotypes. Histograms illustrating continuous variation (two traits in **(b)**; all traits in **(c)** and **(d)**) include gray bars. Insets show representative egg phenotypes from near the ends of each distribution. The data underlying this figure are available at https://doi.org/10.17863/CAM.116928.2.

from one (marking luminance: *P* = 0.16; pattern coverage: *P* = 0.51; Table 1). However, color was best described as two clusters (*n* = 73; *P* = 0.03; Table 1) which corresponded well to human categorization of phenotypes (Fig 2b). These results were largely supported by tests for multimodality (S1 Text §d). Thus, croaking cisticolas exhibited both continuous and categorical variation in traits predicting rejection (Fig 2b).

**iii. Tawny-flanked prinias (hereafter, prinias).** Differences between host and experimental eggs in three traits predicted egg rejection in prinias: color (Estimate = 0.27 ± 0.06SE, $Z_{153}$ = 4.2, $P$ < 0.001), pattern coverage (Estimate = 15.46 ± 4.56SE, $Z_{153}$ = 3.39, $P$ < 0.001), and pattern dispersion (Estimate = 3.38 ± 2.12SE, $Z_{153}$ = 1.59, $P$ = 0.11; Nagelkerke's $R^2$ = 0.36; Table 1). A PAM cluster analysis ($n$ = 254 for egg color, $n$ = 511 for pattern traits) showed no evidence that the number of clusters for any of the three traits differed from one (egg color: $P$ = 0.06, pattern coverage: $P$ = 0.18; pattern dispersion: $P$ = 0.89; Table 1; see S1 Text §c and reference [27] for further evidence against egg color being categorically distributed in prinias). These results were supported by tests for multimodality (S1 Text §d). Thus, prinia eggs showed continuous variation in traits predicting rejection (Fig 2c).

**iv. Red-faced cisticolas.** Differences between host and experimental eggs in two traits predicted rejection in red-faced cisticolas: marking luminance (Estimate = 0.30 ± 0.12SE, $Z_{55}$ = 2.43, $P$ = 0.02) and the number of SIFT features (Estimate = 0.03 ± 0.01SE, $Z_{55}$ = 2.56, $P$ = 0.01; Nagelkerke's $R^2$ = 0.34; Table 1). A PAM cluster analysis ($n$ = 182) showed that for both traits the number of clusters did not differ significantly from one (marking luminance: $P$ = 0.44; number of SIFT features: $P$ = 0.09; Table 1). These results were supported by tests for multimodality (S1 Text §d). Thus, red-faced cisticola eggs showed continuous variation in traits predicting rejection (Fig 2d).

Overall, for each species, the results of cluster analysis (and tests for multimodality; see S1 Text §d) corresponded well with human perception of their egg phenotypes. We also note that for each of the four species, none of the traits predicting rejection were highly correlated with each other (all $R^2$ < 0.23). Thus, each trait can be considered separately when testing for categorical variation.

## b. Rates of Type II errors (accepting foreign eggs)

In field experiments (Section a), the lowest Type II error rate was observed in zitting cisticolas (36 Type II errors out of 90 experiments, 40%; the other three species showed Type II error rates of approximately 50%; Fig 3). While we aimed to choose experimental eggs at random with respect to host phenotype (Methods), this depended on the conspecific eggs available at the time of experiments. This resulted in zitting cisticolas being given experimental eggs matching the

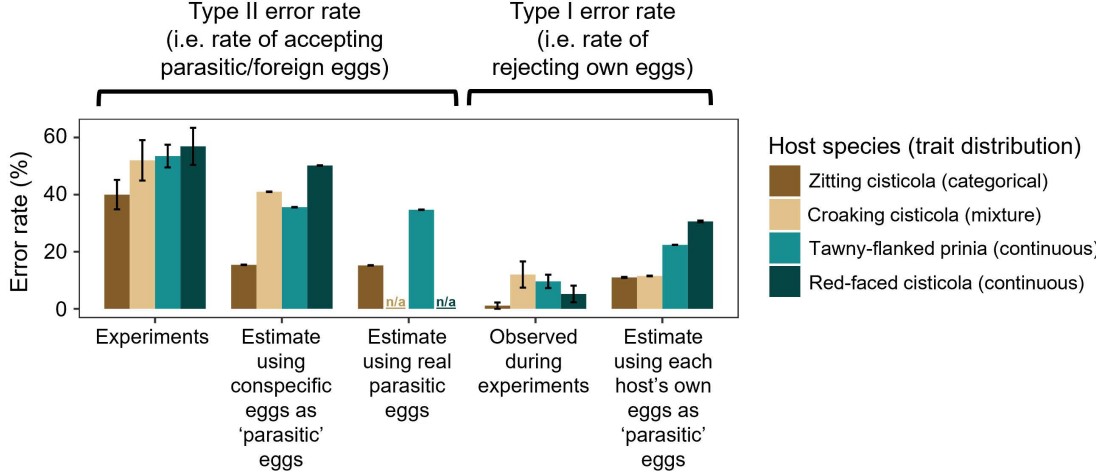

**Fig 3. Observed (from experiments) and estimated (from substituting phenotypic values of host and/or parasitic eggs into the rejection model for each host species) frequencies of Type II and Type I errors in each species.** Estimates using real parasitic eggs were only available for zitting cisticolas and prinias. For estimates of Type II error rates, 1,000 combinations of randomly-selected eggs were used. For estimates of Type I error rates, we generated one estimate per host nest for which there were data for at least two eggs (see Methods for details). Error bars correspond to standard errors. Estimated error rates from rejection models differed from those observed during experiments (see S1 Text §g for discussion). Zitting cisticolas made the fewest errors of both types. The data underlying this figure are available at https://doi.org/10.17863/CAM.116928.2.

category of the host egg on 38/90 occasions (42.2%). This is higher than the 16.7% expected assuming the six categories are equally common, suggesting that our experiments overestimated Type II error rate in zitting cisticolas.

Indeed, when such unavoidable non-randomness in experimental phenotypes was controlled for by randomly substituting known egg phenotypes into rejection models (following reference [28]), we found that zitting cisticolas should make even fewer Type II errors than observed in experiments. We estimated Type II error rates using conspecific host eggs as the "parasitic" treatment for all four host species, by substituting $n = 1,000$ randomly-selected combinations of a host clutch and a conspecific egg from a different clutch into the rejection model. This controls for potential differences in how closely each parasitic lineage mimics its host species, by assuming parasitic eggs have identical trait distributions to host eggs [18]. Zitting cisticolas exhibited the lowest estimated Type II error rate (15.4%); the other three species exhibited rates >35% (Fig 3).

Parasitic egg phenotypes from 2012–2023 were available for prinias [29] and zitting cisticolas (Fig A in S1 Text); thus, we could also estimate Type II error rates with parasitic eggs for these species, again substituting egg phenotypes (here, parasitic eggs alongside host clutches) into rejection models. Zitting cisticolas exhibited a Type II error rate (15.2%) less than half of that for prinias (34.7%; Fig 3). Reassuringly, using host eggs as proxies for parasitic eggs resulted in almost identical estimates of Type II error rate to estimates using real parasitic eggs.

### c. Rates of Type I errors (rejecting own eggs)

In field experiments (Section a), the lowest Type I error frequency was observed in zitting cisticolas (one Type I error out of 90 experiments; Fig 3).

As above for Type II errors, we estimated Type I error rates for each host species. We randomly selected two eggs from a host nest, and estimated the likelihood of rejection given the differences in traits between these eggs and the rejection model for that species. Zitting cisticolas again showed the lowest predicted Type I error frequency (11.0%), though this was only marginally lower than that for croaking cisticolas (S1 Text §g); prinias and red-faced cisticolas showed much higher error rates (Fig 3).

Overall, zitting cisticolas (with categorical traits) made fewer Type II and Type I errors than the other species (with continuous or mixed traits), despite the expected trade-off between the two types of error. In the next section, we use individual-based simulations to test hypotheses for how categorical variation can result in lower rates of both Type I and Type II errors.

### d. Modeling rejection using individual-based simulations

We used individual-based simulations to test whether categorical variation could explain the low rates of both types of errors in zitting cisticolas, and to generalize beyond our empirical system. Effectively, the simulations calculate expected error rates for different host trait distributions, and illustrate two mechanisms by which categorical distributions might result in hosts making fewer Type I and Type II errors.

For each simulation, we generated populations exhibiting either "continuous" trait variation (equivalent to normally-distributed egg traits) or "categorical" trait variation (bimodally-distributed egg traits); hereafter continuous and categorical populations. Populations were generated such that continuous and categorical populations had the same overall trait variance and differed only in the shape of the trait distribution. We estimated error rates and overall success rates in simulations where host and parasitic eggs were drawn from the same underlying trait distributions. This contrasts with typical SDT models, which assume non-identical distributions (see Discussion). Success rate was defined as the proportion of host eggs not subject to either type of error (i.e., eggs that were not in a nest where a parasitic egg was accepted, nor were themselves rejected). In these simulations, Type I errors (rejecting own eggs) are only considered to have an effect in either unparasitized nests or in parasitized nests when a parasitic egg has already been rejected. This is because if a parasitic egg is accepted (i.e., if a Type II error is made), then any Type I errors have no effect since the whole host clutch

is in any case doomed (see Methods and S1 Text §h). This means that the success rate corresponds to the joint probability of avoiding both error types, and is therefore not exactly equal to [1 − (rate of Type I errors + rate of Type II errors)]. Full model descriptions are given in Methods; parameter values are given in Table C in S1 Text.

**i. Threshold-based rejection.** We first simulated continuous and categorical populations (Fig 4a) with threshold-based rejection, where eggs differing from the host phenotype by more than a specified threshold distance are rejected. In general, as the threshold increases, the rate of Type II errors (accepting parasitic eggs) will increase, since more parasitic eggs will fall within the threshold and thus be accepted. Similarly, as the threshold increases, the rate of Type I errors (rejecting own eggs) will decrease, since more of a host's eggs will fall within the threshold and thus be accepted.

At intermediate thresholds, categorical populations experienced greater success than continuous populations (Fig 4b and Fig C in S1 Text). This difference arose because, in categorical populations, rates of Type II errors increase in

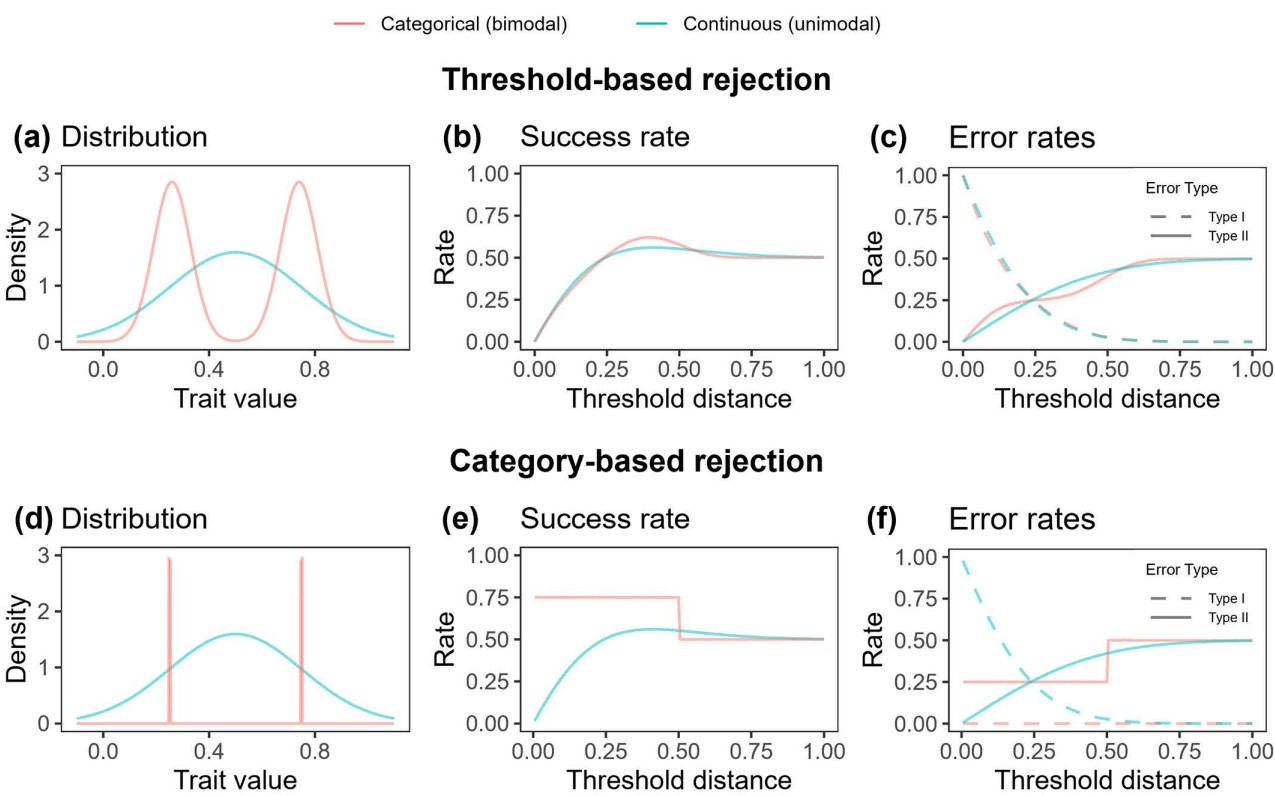

**Fig 4. Estimated rates of Type I errors (rejecting own eggs), Type II errors (accepting parasitic eggs), and overall success rates in simulated populations with parasitism rates of 50%, such that the maximum Type II error rate is 50%.** In panels **(b)**, **(c)**, **(e)**, and **(f)**, "threshold distance" refers to the distance whereby eggs differing from the host phenotype by more than that distance are rejected. Above: Threshold-based rejection of categorical vs. continuous egg phenotypes: **(a)** Simulated categorical (red) and continuous (blue) populations, each with mean trait value 0.5 and standard deviation 0.25. **(b)** Across a range of rejection thresholds, the probability of egg survival in each of the populations. At intermediate thresholds (ca. 0.25–0.55) the success rate in the categorical population is greater than that of the continuous population, due to (c) a reduced rate of Type II errors at these thresholds, without a concurrent increase in Type I error rate. Note that the dashed red and blue lines are virtually identical because, under threshold-based rejection, Type I error rates are unaffected by distribution shape (S1 Text §h). Below: Category-based rejection of categorical egg phenotypes vs. threshold-based rejection of continuous egg phenotypes. **(d)** Simulated categorical (red) and continuous (blue) populations. Here, categorical trait variation is modeled as binary, with no intra-clutch variation, to simulate category-based rejection. **(e)** Category-based rejection occurs in the categorical population at thresholds below 0.5. At all such thresholds, the categorical population experiences greater (or equal) success than the continuous population, due to **(f)** zero Type I errors and a low frequency of Type II errors. Indeed, the Type II error rate of 0.25 in the categorical population is only achievable in the continuous population when thresholds are lower than ~0.25. The simulated data underlying this figure are available at https://doi.org/10.17863/CAM.116928.2.

a stepwise manner with increasing threshold, since the threshold encompasses first one and then both of the peaks of the distribution (solid lines in Fig 4c). To visualize this, consider a host whose eggs lie in the left-hand peak of the bimodal distribution (the following explanation applies similarly to a host whose eggs lie in the right-hand peak). With a low threshold, this host only accepts eggs close to that peak, so all eggs in the right-hand peak are rejected. Increasing the threshold to intermediate values mainly adds eggs from the low-density "valley" between the two peaks, so relatively few additional parasitic eggs fall within the threshold, and the Type II error rate therefore rises only slightly. By contrast, when the threshold is large enough that eggs from the right-hand peak also fall within it, a substantial additional number of parasitic eggs now lies within the threshold, and the Type II error rate increases sharply. Thus, increasing the threshold from low to intermediate values admits few extra parasitic eggs, whereas increasing it from intermediate to high values admits many, producing the stepwise increase in Type II error rate with threshold in categorical populations (solid red line in Fig 4c). This contrasts with continuous populations, where the increase in Type II error rate with threshold is smooth (blue solid line in Fig 4c). As a result, categorical variation allows lower Type II error rates at intermediate thresholds than continuous variation. Meanwhile, Type I error rates remain effectively identical between continuous and categorical populations (dashed lines in Fig 4c; S1 Text §h).

Overall, these simulations demonstrate that distribution shape can influence success rate under threshold-based rejection due to its effect on the proportion of foreign eggs that lie outside a given host's threshold.

**ii. Category-based rejection.** A second potential mechanism for why categorical variation can reduce error rates involves rejection that is not threshold-based, but is instead category-based. A categorical distribution allows for category-based rejection: hosts lay eggs of one category, always accept eggs of that category, and reject all eggs of different categories. We simulated this process for a trait with two categories by generating a truly binary distribution (Fig 4d) with no within-clutch variation, since under category-based rejection, bimodal phenotypes are perceived in a binary manner. We compared error rates between a categorical population exhibiting category-based rejection, and a continuous population (modeled in an identical manner to the continuous population described in (i); see Fig 4a and 4c), exhibiting threshold-based rejection. Category-based rejection does not involve well-defined thresholds, but can be thought of as occurring when the "threshold" is between zero and the distance between the peaks. Under these circumstances, hosts accept eggs of their own category and reject eggs of the other category – i.e., rejection is category-based. When the "threshold" is above the distance between the peaks, all eggs are accepted, and rejection is therefore no longer category-based.

This simulation showed that when eggs are rejected according to their category (i.e., $0 <$ threshold distance $< 0.5$), categorical variation can result in greater success rate than continuous variation (Fig 4e). This is because under category-based rejection, a host's own eggs are always accepted (they always share the host's category), so no Type I errors are made (Fig 4f). Category-based rejection also affects Type II error rates. If there are n equally common egg categories, then the parasitic egg in a parasitized nest matches the host's category with probability $1/n$, and is otherwise rejected. Thus, the probability of a Type II error in a parasitized nest is $1/n$. Across the whole host population, the overall frequency of Type II errors across all nests (both parasitized and unparasitized) is $P/n$, where $P$ is the parasitism rate (Fig 4f). In Fig 4d–4f, for example, $P = 0.5$ and $n = 2$, so the Type II error rate under category-based rejection is 0.25 (i.e., 25% of host nests will involve a Type II error; Fig 4f). Whenever $P/n$ is lower than the Type II error rate in the continuous case (i.e., for sufficiently many categories), the categorical population will exhibit fewer Type I *and* Type II errors than the continuous population, thus escaping the Type I:Type II error trade-off.

Overall, category-based rejection alongside categorical variation can effectively eliminate Type I errors, and concurrently reduce Type II errors relative to continuous variation.

## Discussion

It is widely acknowledged that the extent of variation in biological systems has important consequences, not least in arms races involving models and mimics [17,18,22,25]. In this study, we showed that not only the degree of variation, but

also the shape of trait distributions, can affect how receivers discriminate between models and mimics. Using an African brood parasite–host system, we showed that the host species exhibiting categorical variation (zitting cisticola) made fewer Type II errors (accepting parasitic eggs) and Type I errors (rejecting own eggs) than the other three hosts studied. Individual-based simulations suggested two potential causes of this pattern: one involving threshold-based rejection, and one involving category-based rejection. The latter was directly supported by field data: zitting cisticolas typically rejected eggs of different categories and accepted eggs of the same category to their own. Overall, this study, while necessarily focusing on only four host species, illustrates the role that trait distribution shapes play in influencing ecologically- and evolutionarily-relevant behaviors.

In zitting cisticolas, a strikingly high percentage of variation in rejection (70.0%) was explained by three egg traits (S1 Text §b). By contrast, in other species such as tawny-flanked prinias, less variation in rejection can be explained [19,30]. The traits predicting rejection in zitting cisticolas were categorically distributed (Figs 1a and 2a). While some hosts of other brood parasites appear subjectively to exhibit categorical variation in background color [20,21,31], our results are, to our knowledge, the first to quantify categorical variation in pattern traits used in rejection decisions. Categorical variation in zitting cisticolas was associated with successful recognition of both their own and foreign eggs, a finding inconsistent with the expectation of a Type I:Type II error rate trade-off (see Introduction; S1 Text §e).

How do zitting cisticolas successfully reject foreign eggs, yet still reject few of their own? Individual-based simulations suggested two mechanisms that could drive this effect. First, when rejection thresholds are intermediate, categorical distributions reduce Type II errors relative to continuous distributions, even when trait variance is kept constant. This is because, when variation is categorical, the increase in Type II error rate with increased threshold is step-shaped rather than a smooth curve (Fig 4c). The reduction in Type II error rate could therefore select for more permissive thresholds in species with categorical variation, in turn reducing the frequency of Type I errors [7].

Alternatively, zitting cisticolas may achieve low error rates because categorical variation allows for category-based rejection (Fig 4f). Such category-based decision-making could occur through non-linear perceptual mechanisms, including categorical perception, categorization, and generalization [4,12–14,32]. Rejection in zitting cisticolas appeared almost, but not fully, category-based: with a few exceptions (4/90 trials), eggs of the same category were accepted, and foreign eggs differing from a host clutch in one (or more) categorical trait were rejected. The three categorical traits result in six categories of eggs (Fig 1b). If the six categories are approximately equally frequent (as is largely, though not precisely, the case; Table A in S1 Text), category-based rejection would result in a low Type II error rate (approximately 1/6). Furthermore, hosts would make no Type I errors, since an individual's eggs are all of the same category. Accordingly, hosts may not only coevolve with parasites, but within-species coevolution occurring synergistically between traits and their perception [33] could drive hosts exhibiting categorical variation to evolve category-based rejection, in turn influencing the evolution of egg traits (e.g., evolving more or more discrete categories). Such questions could be best studied in hosts that exhibit categorical traits [20,21,31], or that appear to show categorization-like discrimination behavior [5,14].

Although our study provides evidence for association between categorical variation and low error rates, and mechanisms by which this should occur, such a comparative study cannot directly test for causality. Specifically, we cannot definitively rule out that some other species-specific factors could generate the low error rates seen in zitting cisticolas relative to the other host species. However, species-specific differences in traits such as ability to discriminate small differences, or the degree of inter- and intra-clutch variation, are unlikely to explain why zitting cisticolas exhibit low rates of *both* types of error. This is because varying these factors should concurrently influence the rates of both errors in opposite directions (as discussed in the introduction, with empirical evidence from the present system provided in S1 Text §e; also see [22,25]). Further evidence for the role of categorical variation and category-based perception in reducing error rates might be gathered by empirically testing for these associations in other systems.

Some hosts, such as prinias and red-faced cisticolas, exhibit continuous variation. This indicates either that categorical variation never emerged in these species, or that it arose but was subsequently lost, which may seem surprising given

that categorical variation leads to lower frequencies of Type I and Type II errors. One reason that categorical variation may not inevitably arise, despite its advantages, is that trait distributions are properties of populations, not heritable properties of individuals, so cannot directly respond to selection. Indeed, in a population exhibiting categorical variation, selection on individuals may favor intermediate phenotypes, especially if parasites also exhibit categorical variation and thus do not mimic intermediate host phenotypes. Such selection for intermediate traits might result in continuous variation evolving from categorical variation. Furthermore, hosts may benefit from increased pattern complexity [19,34], since more complex patterns are more difficult to forge. This may involve increasing the number of loci contributing to pattern [19], which may generate continuous variation (see below). Overall, therefore, selection at the individual level may preclude some host populations from exhibiting distributions that would generate increased population-level success against parasites. Future work should aim to test whether such erosion of categorical variation is ultimately inevitable (i.e. categorical variation simply has not *yet* been lost in zitting cisticolas), or whether it might be prevented by ecological or genetic factors such as specific genetic architectures.

Continuous and categorical variation are likely underpinned by different genetic architectures. Categorical variation typically results from genetic variation at few loci, each with large phenotypic effects, rather than variation at many small-effect loci [35]. For instance, two alternative alleles at a single locus of large effect influencing an egg trait could lead to a bimodal distribution for that trait, as seen in zitting cisticolas. By contrast, the presence of numerous small-effect epistatic modifier loci can result in trait variation becoming more continuously distributed [15]. Such putative variation in genetic architectures underpinning cisticolid egg phenotypes in closely related species (all confamilial; three of four congeneric) would be further evidence of the power of coevolution to drive diversification within and between species [17,36].

More broadly, our results extend SDT in biology. SDT largely arose in psychology, in the context of receivers discriminating signal from noise [8,9,37], but biological recognition systems (where receivers discriminate between different signals) are now often analyzed in terms of SDT [38]. SDT predicts trade-offs between Type I and Type II errors in conspecific and heterospecific recognition, with implications for speciation and mate choice [39], in discriminating models from mimics [40,41], and in coevolution between cognitive systems and signals [42]. Our findings complement this literature, which typically models the two entities being discriminated (signal versus noise, or model versus mimic) as having continuous distributions differing in mean and/or standard deviation [9,24,38]. Indeed, SDT considers cases where the two distributions are identical as comprising no discriminative information, such that any decision rule performs at chance [8,9]. By contrast, we show that category-based discrimination (when receivers form discrete "own-category" versus "other-category" templates) can result in low rates of Type I and Type II errors, even if population-level model and mimic distributions are identical. This extends the classical SDT framework by highlighting that categorical distributions change which error rates are achievable. Future work could examine continuous and categorical distributions, both empirically and mathematically, to understand how decision-making could be optimized under different combinations of variation and perception.

Such an understanding would be especially relevant to self-recognition systems beyond the avian host-parasite systems studied here, including insect host–parasite systems with chemical mimicry, and immune recognition of pathogenic antigens versus host (self) antigens. Trade-offs between errors are crucial in all these systems: for instance, in the immune system, Type II errors lead to infection while Type I errors cause autoimmunity [17]. Importantly, in self-recognition systems, even though host distributions may contain huge inter-individual variation that is partially or fully mimicked by parasites, each individual host possesses a (typically learnt) template of its own traits [17]. This allows hosts to successfully identify even parasites that possess traits indistinguishable from those of other hosts in the population [18]. Such discrimination could be aided by recognition systems similar to category-based recognition in zitting cisticolas. For instance, in vertebrate immune systems, there is considerable Major Histocompatibility Complex (MHC) variation even within individuals, and individual MHC molecules can bind thousands of different peptides with similar structures, likely resulting in particular categories of peptides being more easily recognized than others [17,43]. Together, these parallels motivate quantifying trait distributions—and how they are perceived—across recognition systems confronted by mimetic antagonists.

## Methods

### a. Ethics statement

Fieldwork was conducted under Permit Number DNPW/8/27/1 (Department of National Parks and Wildlife, Zambia), with ethical approval from the local Animal Welfare and Ethical Review Body (AWERB) that oversees animal research undertaken at the University of Cambridge.

### b. Field experiments and quantifying egg phenotypes

We studied wild populations of the cuckoo finch *Anomalospiza imberbis* and four of its hosts: zitting cisticola *Cisticola juncidis*, croaking cisticola *C. natalensis*, tawny-flanked prinia *Prinia subflava* (hereafter, prinia), and red-faced cisticola *C. erythrops*. Field experiments were conducted on and around Semahwa Farm (ca. 16.74°S, 26.90°E) in the Choma District of southern Zambia during wet seasons (January–April) in 2018–2020 and 2022–2023. Each experiment was conducted at a different host nest. Experiments were conducted on prinias (2022, 2023; $n = 157$), zitting cisticolas (2018–2020 and 2023, $n = 90$), red-faced cisticolas (2019, 2020, 2022, 2023, $n = 58$), and croaking cisticolas (2019, 2020, 2022, 2023, $n = 50$).

Experiments were carried out as in reference [44]; one egg from a host nest was replaced with a conspecific egg from a different nest (the "experimental egg") and the entire host clutch was photographed alongside the experimental egg. Eggs were placed in host nests approximately at random to simulate the full range of variation to which hosts are exposed, unlike previous experiments in which experimental eggs were well-matched to the host clutch [19,25,29]. Based on previous work [19,25,28,29,44], nests were monitored for four days: if an experimental egg was not rejected within four days, it was treated as accepted. Unlike the other hosts, which sometimes took several days to reject eggs, as also observed in previous studies [19,25,28,29,44], zitting cisticolas almost always rejected within 24 hours (occasionally rejection occurred up to 48 hours, but foreign eggs were never rejected after 48 hours).

We quantified egg phenotypes (color and pattern) in the experiments above, and for natural clutches containing host and/or parasitic eggs in 2012–2023, using methods [45–53] described in reference [29] (see S1 Text §a for full details of color and pattern quantification). We used principal component analysis to determine the main axes of variation in egg color. Principal components (or analogous dimensionality-reduction measures) can reduce dimensionality and account for collinearity in cone catch values, and have therefore been used to quantify color according to organisms' visual systems in studies on a range of taxa [28,54–57].

To determine which traits predict rejection for each host species, we used logistic regression (function glm) in R [58] to model rejection behavior. For all models, the predictors were the absolute differences between the experimental egg and the mean host egg in the nest for each of the traits described in S1 Text §a, and the response was whether the experimental egg was rejected. For each species except for croaking cisticolas, the full model contained all traits, namely the difference between the experimental egg and the host clutch average in color (measured in JNDs), luminance (measured in JNDs), principal marking size, marking size variation, total contrast, pattern coverage, pattern dispersion, number of SIFT features, mean feature size (referred to as "marking size" in Results), and pattern luminance. Croaking cisticolas sometimes lay immaculate eggs (Fig 2b). Although other "pattern" measures can still be calculated for immaculate eggs (see, e.g., [18,22]), marking luminance cannot be quantified for immaculate eggs. Therefore, we ran two models for croaking cisticolas. For one ($n = 38$), we only included experiments in which neither the host clutch nor the experimental egg was immaculate, and included all traits (except for the number of SIFT features, since this was highly correlated with pattern coverage; Pearson's $r = 0.64$) in the full model. For the other ($n = 50$), we included all experiments, but excluded marking luminance (since immaculate eggs lacked values for this trait) and number of SIFT features from the full model. Results from these models are reported in Results, but see S1 Text §b for full details on the two models for croaking cisticolas.

We used the R package *MuMIn* [59] to compare models based on Akaike Information Criteria AICc [60]. The best model was selected as the simplest model from the subset of models within 2 AIC of the model with the lowest AIC value. We used Nagelkerke's $R^2$ (package *rsq* in R) to calculate percentage of variation explained by each model.

### c. Testing for categorical variation

We used PAM cluster analyses (*pamk* function in the *fpc* package in R [61]) to test whether each of the color and pattern traits that predicted (i.e., were associated with) egg rejection in each species comprised more than one cluster (see S1 Text §c for details), and how many clusters (from 1 to 10) best predicted the data. We used Duda–Hart tests (also in the *fpc* package) to determine whether the optimal number of clusters differed significantly from one [61]. By assigning individual eggs to clusters, we could compare the output of the cluster analysis with human categorization of those traits that were categorically distributed. However, while cluster analysis provides information about the grouping of data, it is not a direct test of multimodality (i.e., testing whether a distribution contains two or more modes, that therefore indicate multiple categories). We therefore confirmed categorical variation in those traits for which the optimal number of clusters was greater than one by using two tests for multimodality. First, we used Hartigans' dip test [62] (package *diptest* in R) which tests whether the number of modes in a distribution is greater than one. We then used Silverman's tests [63] (package *Multimode* in R), which test for more than n modes (i.e., replicating the Hartigans' dip test when $n = 1$, but unlike Hartigans' dip tests, we also used Silverman's tests to test for more than two modes). For $n = 1$, we used the Hall–York calibration [64], which improves the accuracy of the test when testing for more than one mode. We then used Silverman's tests with $n = 2$ to determine whether distributions comprised more than two modes. In no case did any distribution comprise more than two modes, so further testing (e.g., for more than three modes) was unnecessary. For all Silverman's tests, we used 100 bootstrap replications and calculated the critical bandwidth and its associated *p*-value.

We report results from Hartigans' dip tests and Silverman's tests in S1 Text §d, rather than in the main text, because these generated identical conclusions to cluster analyses (with the minor exception of some specific differences for some traits in croaking cisticolas; Table B in S1 Text). Moreover, results from cluster analyses are easily visualized and compared to human categorizations: cluster analysis allows assignment of eggs to specific clusters (categories), facilitating visualization of which eggs fall into which clusters and comparison to human classification (e.g., Fig 2). This is not possible for direct tests of multimodality, since eggs cannot be "assigned" to a particular mode.

### d. Estimating error rates for each species

**i. Estimating Type II error rates for each species.** Our experiments provided estimates of Type II error rates (accepting parasitic eggs) for each species – these "observed" rates were simply the proportion of experiments for which individuals of each host species accepted the conspecific egg placed in their nest to simulate parasitism. However, we also estimated rejection rates based on the rejection model of each species to control for the possibility that, by chance, different species were given differently-well-matched eggs across experiments (c.f. [28,44]). While we aimed to randomly choose experimental eggs with respect to host clutch phenotypes, our choice of experimental eggs was dependent on the host eggs present at the time of experimental manipulation that could be used as experimental eggs. Thus, as with all egg rejection experiments using conspecific or parasitic eggs, we could not ensure that all hosts were given equally difficult tasks [28,44]. If we had inadvertently given one species more difficult tasks than another, this would artificially increase the observed Type II error rate of the first species relative to the second, without necessarily indicating that the first is actually poorer at discriminating its eggs from those of its parasite. The solution to this problem is to estimate rejection rates based on all phenotypic data available [28,44]. Note that we did not use statistical analyses here (following previous work [44]) due to sample sizes being arbitrary. Results can be visually compared in Fig 3.

Estimation of rejection rates involved parametrizing the rejection model with data from real eggs to estimate the likelihood of rejection based on random laying by the parasite. Parasites appear to lay at random with respect to host phenotype in this system [26,28,44].

For prinias and zitting cisticolas, data on parasitic eggs from 2012 to 2023 were available such that rejection rates of real parasitic phenotypes could be estimated. Each parasitic egg analyzed was laid in a different host nest (where more than one parasitic egg was laid in the same host nest, which is common [65], we randomly selected one for further analysis). This resulted in a dataset of $n = 129$ eggs laid in prinia nests (published in reference [29]) and $n = 34$ eggs laid in zitting cisticola nests (see S1 Text §f). We randomly selected one host egg from a clutch to represent the phenotype of the clutch. We then randomly selected a parasitic egg, and calculated the probability that it would be accepted according to the rejection model for that species. We generated 1,000 such combinations, with resulting estimates of Type II error rate shown in Fig 3.

The estimated Type II error rate using parasitic eggs is not necessarily comparable between species, since different hosts may be subjected to differing degrees of mimetic accuracy and phenotype distributions by the parasite lineages (known as "gentes") that specialize on each host. We therefore also estimated parasitism events using host eggs as a proxy for parasitic eggs. This assumes that the distribution of parasitic phenotypes is identical to the distribution of host phenotypes. Using host eggs as proxies for parasitic eggs does not necessarily correspond to true biological reality (since in at least two of the host species studied here, there are consistent differences between parasitic and host eggs [19,28,29,34]), but it allows us to compare host species' propensity for Type II errors controlling for potential differences in the accuracy of parasitic mimicry for each host. As with the estimated parasitism events described above, we randomly selected one host egg from a clutch. We then randomly selected another host egg from a different clutch, as a proxy for a parasitic egg, and calculated the probability of rejection of this egg based on the rejection model for that species. We generated 1,000 such combinations, with resulting estimates of Type II error rate shown in Fig 3.

Estimates of Type II error rates calculated as above are provided in the main text and summarized in Fig 3. These estimated Type II error rates were lower than observed in experiments (Fig 3). This is likely due to us inadvertently providing birds with slightly more difficult decisions (i.e., more closely matched eggs) than the intended random assignment. As discussed above, the assignment of "experimental" eggs to host clutches depended on the host eggs available to us at the time, and therefore it was not possible to make this assignment truly random. Furthermore, the bias could result because, in order to test effects of pattern differences, we needed to provide good color matches (and vice versa); this likely resulted in providing a greater proportion of good color and pattern matches than a truly random assignment would generate [18].

**ii. Estimating Type I error rates for each species.**  We estimated Type I error rates by generating combinations of eggs, each from the same clutch, quantifying the differences between them and estimating the probability that one of the eggs would be rejected according to that host's rejection model. Only a subset of host clutches had reflectance spectra measured for multiple eggs within the same clutch ($n = 73$, 28, 21, and 30 for prinias, zitting cisticolas, red-faced cisticolas, and croaking cisticolas, respectively) because reflectance spectra were collected indoors at the field site base, not in the field. We avoided removing multiple eggs for spectrophotometry because this might influence rejection behavior. To incorporate color differences in our error rate estimations, we therefore randomly assigned to each pair of eggs one of the values for intra-clutch color difference from the subset of nests for which multiple eggs were measured. Because the JNDs were randomly assigned to comparisons, and because pattern and color traits are uncorrelated in hosts [19,44] (also see Results), this should not introduce any bias into these estimations.

Estimates of Type I error rates are provided in the main text and summarized in Fig 3. These estimates predicted higher rates of Type I errors than observed during field experiments, except for croaking cisticolas (see main text; also see S1 Text §g for a potential explanation for this relating to croaking cisticolas possessing large bills [66]). It may therefore be that hosts use other (unmeasured) cues to recognize their own eggs and thus avoid rejecting them more than predicted

 

by rejection models. Another possible (and not mutually exclusive) explanation is that a foreign egg placed in a nest draws attention away from any of the host's own eggs that are somewhat dissimilar and might otherwise be rejected.

We calculated standard errors (SEs) for all observations and estimates of Type I and Type II errors (Fig 3). For observations, SEs were estimated using the formula

$$SE = \sqrt{\frac{p(1-p)}{n}}$$

where $p$ is the frequency of errors and $n$ is the sample size. For estimates of Type II error rates, we arbitrarily chose a sample size of 1,000 combinations of eggs, and calculated the mean error rate. We estimated SEs of these means using bootstrapping. To do this, we randomly sampled (with replacement) 100 combinations from the 1,000, and calculated the mean error rate. We repeated this 100 times and calculated the SE of the 100 estimated error rates. For estimates of Type I error rates, which were not based on 1,000 replicates, we calculated SEs directly without bootstrapping.

### e. Individual-based simulations

We used individual-based simulations to test how categorical variation versus continuous variation could influence rates of errors and thus success in host populations. The simulations were written in Python 3.11 using the standard Math, Random, and NumPy libraries.

**i. General model description.** Each individual-based simulation involves a host and a brood parasite population, where, for simplicity, each population is haploid. Because these are not evolutionary simulations, the ploidy of the population does not influence results. In each replicate, all nests have a nesting attempt, a potential parasitism event, and a series of host decision processes for all eggs within the nest. There are $N_{nest}$ nesting sites and each of these is occupied by a host parent, who lays a clutch size of $N_{egg,host}$ eggs (constant across all nests). Each nest has a probability of being parasitized, $p_{parasitized}$, and if the nest is parasitized, the parasite lays $N_{egg,para}$ eggs in it. For all simulations $N_{egg,host} = 3$ and $N_{egg,para} = 1$. Although cuckoo finches often, though not always, lay more than one egg in each host nest (perhaps because increasing the proportion of parasitic eggs in a host nest reduces the likelihood of the host rejecting the parasitic egg or eggs [65]), we keep $N_{egg,para}$ constant to ensure all parasitized host nests are comparable, and because these simulations are intended to test the role of trait distribution shape in a generalizable way, rather than encapsulating all aspects of natural history in the cuckoo finch system.

Each simulated host and parasite population possesses one egg trait, distributed on a one-dimensional space. The host uses only this single trait in deciding whether to reject eggs. Egg traits are products of the parental genotype.

The survival of host eggs ("success") is entirely based on rejection decisions (i.e., food availability, predation, offspring quality, and other potential complications are excluded for simplicity), and is hierarchical based on whether a parasitic egg is accepted. This means that, if a parasitic egg is accepted (i.e., a Type II error occurs), no host offspring are produced, as is nearly always the case in the cuckoo finch system [44]. By contrast, if a parasitic egg is rejected or the nest is not parasitized, the host then inspects each host egg. If a host egg is rejected, a Type I error occurs; if a host egg is accepted, the resulting host offspring survives.

Hosts and parasites are defined as having the same genotype distribution, which underlies the egg trait involved in rejection. The genotypic values corresponding to the trait are distributed either continuously or categorically. Specific details for how these are generated depend on the particular distribution being created, and are therefore detailed below.

Host egg phenotypes are generated from the genotype values using a layer of normal distribution to represent the within-clutch variation, centered at the genotype of the focal host,

$$E_{host,pheno,i,j} \sim Norm(H_{geno,i},\ \sigma_{host})$$

$$E_{\text{para,pheno},i} = P_{\text{geno},i}$$

where $E_{\text{host,pheno},i,j}$ is the phenotype of the focal host egg, $E_{\text{para,pheno},i}$ is the phenotype of the focal parasitic egg, $i$ is the index of the host, $i \in \{0, \ N_{\text{nest}}\}$, and $j$ is the index of the host egg, $j \in \{0, N_{\text{egg,host}}\}$. $H_{\text{geno},i}$ and $P_{\text{geno},i}$ are the genotypes of the host and parasite individual at the focal nest, and $\sigma_{host}$ is the degree of host within-clutch variation, which is assumed to be a constant for the entire host population. The parasite population does not require within-clutch variation because in all models, parasites only lay one egg (i.e., $N_{\text{egg,para}} = 1$).

The decision to accept or reject is based on the threshold distance of the host parent ($H_{\text{thrs}}$). Specifically, two thresholds are created based on the genotype of the host and the threshold distance: an upper threshold, $H_{\text{geno},i} + H_{\text{thrs},i}$, and a lower threshold, $H_{\text{geno},i} - H_{\text{thrs},i}$. If an egg in the host nest (either laid by the host itself or laid by a parasite) has a trait value between both thresholds, then it is accepted by the host parent. The parameters $H_{\text{geno},i}$ and $P_{\text{geno},\ i}$ take different values in each model (see S1 Text §h for details). We note that while, for simplicity, we have used strict thresholds (i.e., where all eggs falling outside the threshold will be rejected, and all eggs falling within the threshold will be accepted), thresholds are likely to be less strict in nature, such that some eggs falling outside thresholds will be accepted and vice versa.

We conduct 1,000 replicates, each with 1,000 nests, so that in total $10^6$ independent nesting events are recorded for each combination of parameters. A full summary of parameter names, values, and description is given in Table C in S1 Text.

**ii. Simulating threshold-based rejection for hosts with various shapes of continuous and categorical distributions.** This model (see Results Section d) requires a continuous and categorical population, that each reject eggs according to thresholds (in contrast to the category-based rejection model below, where the categorical population does not reject according to thresholds). We generated these populations from normal distributions.

Specifically,

$$H_{\text{geno}}, \ P_{\text{geno}} \sim \text{Norm}\left(0.5, \ \sigma_{\text{cont}}\right),$$

for the continuous case, and

$$\begin{cases} \Pr = \frac{1}{2}; H_{\text{geno}}, \ P_{\text{geno}} \sim \text{Norm}(0.5 - d_{\text{cat}}, \sigma_{\text{cat}}) \\ \Pr = \frac{1}{2}; H_{\text{geno}}, \ P_{\text{geno}} \sim \text{Norm}(0.5 + d_{\text{cat}}, \sigma_{\text{cat}}) \end{cases},$$

for the categorical case. We ensured that the overall standard deviation is identical for both continuous and categorical cases by applying the formula $\sigma_{\text{cont}}^2 = d_{\text{cat}}^2 + \sigma_{\text{cat}}^2$.

We varied the shape of the categorical distribution (from highly bimodal to almost unimodal) by changing the standard deviation $\sigma$ of each of the two normal distributions making up the bimodal distribution (Fig Ca–e in S1 Text). By doing so, and by changing the half distance $d$ between the two peaks, we ensured that the overall standard deviation of the unimodal and bimodal distributions were always equal. We tested how a range of rejection thresholds impact the frequency of Type I and Type II errors for each distribution. One case is provided in the main text (Fig 4a–4c, duplicated in Figs Cd, Ci, Cn, and Cs in S1 Text); the remaining panels in Fig C in S1 Text illustrate how variation in the shape of the categorical distribution (from highly bimodal to almost unimodal) affects error rates compared to a comparable continuous distribution. Furthermore, panels k–t in Fig C within S1 Text illustrate how Type I and Type II error rates vary with threshold distance for each of the parameter combinations, and thus contribute to the egg success rate (Fig Cf–j in S1 Text). Within-clutch variation $\sigma_{\text{host}}$ was set to 0.25 and the probability of parasitism $p_{\text{parasitized}}$ was 0.5.

These individual-based simulations are highly simplified (e.g., simulating only one egg trait, keeping within-clutch variation constant, and keeping parasitism rate constant). All of these factors should affect success rate (e.g., more egg traits should increase the likelihood of mismatches in at least one trait between host and parasitic eggs; increased within-clutch

variation should increase the frequency of Type I errors and thus favor higher rejection thresholds; increased parasitism rate should increase the probability of making Type II errors, and thus favor lower rejection thresholds); however, these impacts will be similar for both continuous and categorical distributions, so are not explored further here. Moreover, we only estimate error rates and not fitness consequences, since the latter depend on the relative cost to a host of being parasitized versus rejecting one or more of its own eggs (which in turn may depend on various ecological factors that are system-specific). More importantly, the individual-based simulations draw both host and parasitic eggs from the same distributions. This is justified by the empirical data and estimation of Type II errors reported in the main text, but neverthe-less even highly mimetic parasites such as cuckoo finches are unlikely to have identical trait distributions to their hosts in nature. This may be because of evolutionary lag meaning that parasites are yet to "catch up" to hosts [19], weak selec-tion on parasites due to imperfect host perception [29], chase-away selection driving hosts away from parasites [34,67], bet-hedging by parasites [68], the inability of parasites to produce certain phenotypes that hosts can produce [26], or even host perceptual biases such as Weber's Law driving parasites to exhibit specific imperfections [6]. Such complexi-ties mean that the individual-based simulations presented here do not necessarily reflect true biological reality. Instead, they illustrate in a generalizable way that trait distributions can influence error rates in ways not fully encompassed by the expectation of a trade-off between Type I and Type II errors.

 **iii.  Simulating category-based rejection.**  A categorical distribution allows for the possibility of category-based egg recognition, such that hosts lay eggs of one category, and always accept eggs of that category, but reject eggs of the other category (or categories). As discussed in the main text, this process can be simulated for a single egg trait with two categories by generating a truly binary distribution, since under categorical rejection egg phenotypes with two categories would be perceived in a binary manner. We therefore set standard deviation of each peak $\sigma_{cat} = 0$ and within-clutch variation $\sigma_{host} = 0$. We compared the Type I and Type II error frequency under this distribution with the error frequencies of a comparable continuous population (i.e., one with the same overall standard deviation as the categorical population). Threshold was varied as before; category-based rejection occurs in the categorical population when the threshold is set between zero and the distance between the peaks, $d$. Results of this simulation are provided in Fig 4d–4f.

## Supporting information

**S1 Text. Includes supplementary methodological details, results, discussion, and the following figures and tables. Fig A.** Histograms illustrating distributions of traits in the cuckoo finch gens parasitizing zitting cisticolas. The three traits illustrated are those that predict rejection in zitting cisticolas: color, marking luminance, and mean feature size. Insets show representative egg phenotypes from near the ends of each distribution. The data underlying this figure are available at https://doi.org/10.17863/CAM.116928.2. **Fig B.** Pattern differences between cuckoo finch ($n = 129$) and prinia ($n = 371$) eggs in the present dataset. Differences were similar to those found in reference [44] on a dataset of prinia and cuckoo finch eggs measured at the same study site in 2007–2009. $P$-values refer to Wilcoxon rank-sum tests. Whiskers extend to the most extreme value within 1.5 * IQR (inter-quartile range) of the corresponding hinge of the box. Hinges correspond to first and third quartiles; the IQR is the distance between the first and third quartiles. The data underlying this figure are available at https://doi.org/10.17863/CAM.116928.2. **Fig C.** The summary of simulated results with various shapes of categorical distributions. (a–e) The genotype distributions, where increasing distance between the two categor-ical peaks results in decreased width of each peak (such that the overall host genotypic variation is constant). From left to right, the categorical distribution becomes more discrete, resulting in greater differences in (f–j) success rate, (k–o) Type I error rate, and (p–t) Type II error rate between the respective categorical and continuous populations. In many instances, the categorical population exhibits greater success than the continuous population. Note that panels d, i, n, and s illustrate the same results as shown in Fig 4a-c, and are included here for completeness. The simulated data underlying this figure are available at https://doi.org/10.17863/CAM.116928.2. **Table A.** Frequencies of clutches of each category in zitting cisticolas, according to the human eye and PAM clustering. **Table B.** Comparison of results from cluster analyses and

direct tests for multimodality. Conclusions from each test are provided alongside P-values (see main text and S1 Text §d for details including test statistics and sample sizes). **Table C.** Values and descriptions of parameters in individual-based simulations.
(DOCX)

## Acknowledgments

In Zambia, we thank many people who assisted with fieldwork, including Silky Hamama, Sanigo Mwanza, and Onest and Oscar Siakwasia; the Sejani, Nicolle, Bruce-Miller, Duckett, and Greenshields families for hospitality and allowing us to conduct fieldwork on their properties; Moses Chibesa, Lackson Chama, and Stanford Siachoono at Copperbelt University for their support; and the Department of National Parks and Wildlife for Research for Permit Number DNPW/8/27/1, under which research was conducted. We thank Kaspar Delhey, Mairenn Attwood, and Gabriel Jamie for comments or discussion, and Jolyon Troscianko, Will Feeney, Martin Stevens, and Wenfei Tong for contributing to data collection in 2012–2014. We also thank two anonymous reviewers and the academic editor (Michael D. Jennions) for very helpful comments on a previous version. Research reported in this article was supported and made possible by the Max Planck–University of Cape Town Centre for Behaviour and Coevolution in collaboration with the FitzPatrick Institute of African Ornithology at the University of Cape Town and the Max Planck Society.

## Author contributions

**Conceptualization:** Tanmay Dixit, Claire N. Spottiswoode.

**Data curation:** Tanmay Dixit, Ming Liu, Jana M. Riederer.

**Formal analysis:** Tanmay Dixit, Ming Liu, Jana M. Riederer.

**Funding acquisition:** Tanmay Dixit, Claire N. Spottiswoode.

**Investigation:** Tanmay Dixit, Ming Liu, Jana M. Riederer, Jonah M. Walker, Cameron J. Blair, Jess Lund, Claire N. Spottiswoode.

**Methodology:** Tanmay Dixit, Ming Liu, Jana M. Riederer, Jess Lund, Claire N. Spottiswoode.

**Project administration:** Tanmay Dixit, Collins Moya, Claire N. Spottiswoode.

**Resources:** Tanmay Dixit, Collins Moya, Claire N. Spottiswoode.

**Software:** Tanmay Dixit, Ming Liu.

**Supervision:** Claire N. Spottiswoode.

**Validation:** Tanmay Dixit, Ming Liu, Jana M. Riederer.

**Visualization:** Tanmay Dixit, Ming Liu.

**Writing – original draft:** Tanmay Dixit.

**Writing – review & editing:** Tanmay Dixit, Ming Liu, Jana M. Riederer, Jonah M. Walker, Cameron J. Blair, Jess Lund, Claire N. Spottiswoode.

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
