## [Editor Report · Decision Letter 0]

10 Dec 2025

Dear Dr Dixit,

Thank you for submitting your manuscript entitled "Categorical versus continuous signatures of identity in host–parasite coevolutionary arms races" for consideration as a Short Report by PLOS Biology.

Thank you for addressing the editorial suggestions that we proposed when you submitted the original version of your manuscript. As discussed previously, we will now send your paper out for external peer review.

Once your full submission is complete, your paper will undergo a series of checks in preparation for peer review. After your manuscript has passed the checks it will be sent out for review. To provide the metadata for your submission, please Login to Editorial Manager (https://www.editorialmanager.com/pbiology) within two working days, i.e. by Dec 12 2025 11:59PM.

Kind regards,

Taylor

Taylor Hart, PhD,

Associate Editor

PLOS Biology

thart@plos.org

---

## [Decision Letter · Decision Letter 1]

21 Jan 2026

Dear Dr Dixit,

Thank you for your patience while your manuscript "Categorical versus continuous signatures of identity in host–parasite coevolutionary arms races" was peer-reviewed at PLOS Biology. It has now been evaluated by the PLOS Biology editors, an Academic Editor with relevant expertise, and by several independent reviewers.

Based on the reviews, we are likely to accept this manuscript for publication, provided you satisfactorily address the remaining points raised by the reviewers and the academic editor (copied after the reviewer reports). Please also make sure to address the following data and other policy-related requests.

IMPORTANT: please ensure that your next revision implements the following editorial points:

------------------

**Title:

To emphasize the main findings of your study, we suggest an alternative title for your paper. Is the following acceptable to you?

"Categorical identity signatures can reduce host error rates during brood parasitism"

**Financial disclosure statement:

-- Please include in your statement the answer to the following question:

"Did the sponsors or funders play any role in the study design, data collection and analysis, decision to publish, or preparation of the manuscript?"

**Ethics:

-- Please provide an approval number for AWERB, if applicable. Please also place the ethics information in a new, independent, and first sub-section of the Materials and Methods section called "Ethics statement".

**Data:

-- Thank you for uploading your data and code to the Cambridge data repository. Please now make them available so that we can assess whether we need any additional information, prior to formal acceptance of your paper.

-- Please note that we require that all of the numerical values underlying the plots in the main and supplementary figures are provided either in supplementary excel files or in the online repository.

-- Please cite the location of the data clearly in all relevant main and supplementary Figure legends, e.g. “The data underlying this Figure can be found in S1 Data” or “The data underlying this Figure can be found in https://doi.org/10.5281/zenodo.XXXXX”

**Supplement:

We see that you have included supplemental methods, results, tables, and figures in your supporting information document. Because documents of this kind are not proofread and are rarely examined by readers, our preference is that these items are integrated into the main paper as much as possible.

-- Please upload the supplementary figures separately as supporting information files. Please include all supplementary item titles and legends at the end of the main text file, as well as supplementary tables.

-- As references that are only found in the supplement will not be included in citation records, please ensure that all references in the supplement are also included in the main references list.

-- Supplementary files (e.g., excel). Please ensure that all data files are uploaded as 'Supporting Information' and are invariably referred to (in the manuscript, figure legends, and the Description field when uploading your files) using the following format verbatim: S1 Data, S2 Data, etc. Multiple panels of a single or even several figures can be included as multiple sheets in one excel file that is saved using exactly the following convention: S1_Data.xlsx (using an underscore).

**Code availability:

-- Per journal policy, if you have generated any custom code during the course of this investigation, we require that you make it available without restrictions. Please ensure that the code is sufficiently well documented and reusable, and that your Data Statement in the Editorial Manager submission system accurately describes where your code can be found.

-----------------

We expect to receive your revised manuscript within two weeks.

*Published Peer Review History*

*Press*

Sincerely,

Taylor

Taylor Hart, PhD,

Associate Editor

thart@plos.org

PLOS Biology

Reviewer remarks:

Reviewer #1: This manuscript presents important new insights into the coevolutionary arms race between brood parasitic birds and their host species, combining empirical field studies with individual-based simulations. A primary conclusion of the paper is that there is not necessarily an inherent trade-off between optimizing Type 1 and Type 2 errors (that is, minimizing the rejection of the host's own eggs while at the same time maximizing the rejection of the parasite's eggs) as is usually predicted by signal detection theory. This novel conclusion is not only significant for understanding the coevolution of brood parasites and their hosts, but also more broadly applicable to other systems in which there is a potential trade-off between recognition of self and non-self. I expect this may become a widely-cited paper.

The empirical data and models presented here strongly support the manuscript's main conclusions. The manuscript is exceptionally well written. Truth be told, I do not have any major concerns with methods, analyses, or the interpretation of results. The following are some relatively minor suggestions for improving an already strong paper (in order as they occur in the manuscript):

l. 79 - The term "multimodal" is appropriate to use to describe the distribution of potentially categorical traits. However, this term is now also commonly used in the context of signaling and animal communication to refer to the involvement of more than one sensory modality in a single functional signal (e.g., a bird simultaneously vocalizing while performing a visual display). So I'm (slightly) concerned that the sentence "Such traits are 'multimodel': their distribution included more than one modal value" might be initially misleading to some readers. Perhaps this (slight) concern of mine would be addressed by rewriting this sentence as "Such traits are 'multimodal,' meaning that the distribution of trait values has more than one peak."

l. 110 - Given that the previous two sentences are referring to hypotheticals, I think it would be good to explicitly state that the negative correlation between consistency and distinctiveness in eggs is an empirical observation. Something like "However, empirical observations have shown that consistency and distinctiveness in egg appearance…"

Fig 1 - This is a very useful figure! But I found myself wanting to see a similar array of eggs for the croaking cisticola, given that its eggs are neither apparently categorical (as illustrated for the zitting cisticola in panel a) nor apparently continuous (as illustrated for the tawny-flanked prinia in panel b). Perhaps the combination of categorical and continuous traits would be less visually obvious (or harder to pull out from looking at the images, but wouldn't that help make the point?

l. 164 - I realize that PAM cluster analysis is explained in the Methods, but given that this isn't a widely used method, I think it would be good to spell the name out when first mentioned here in the text since the acronym alone seems like jargon.

ll. 414-426 - Two point about this paragraph: First, after reading this several times I wasn't completely sure if the authors are suggesting that categorical variation of traits is the ancestral state with continuous variation inherently being the derived state. If that's the case, it would be good to state such more explicitly. Second, if categorical variation is ancestral, I wonder if it isn't equally important to ask why zitting cistolas have maintained categorical variation in the face of the selective factors that lead to continuous variation as described in this paragraph. I certainly may be misunderstanding something here, but this was the one point in the paper where I didn't fully follow the argument being made, so perhaps this could be clarified.

l. 581 -I understand that the terms "gen" or "gente" are used to refer to host-adapted lineages in the brood parasitism literature. I don't think the term is widely used in other areas of biology (although it is more so in anthropology), so it stands out here a some field-specific jargon. Maybe this should be described more completely here for the non-specialist. (This also appears in one or two places in the supplementary materials.)

Reviewer #2: This is an exciting manuscript that shows how far the field has come since the first work to codify egg rejection, thresholds, and error rates by Davies et al. 1996 (Proc B). Sophisticated measures of egg coloration and patterning are combined with statistical approaches to demonstrate continuous versus clustered categorical 'signatures' in several host species of the brood parasitic African cuckoo finch. By embedding these measures within the results of field experiments, these data therefore not only confirm our human-vision 'hunches' of egg phenotypes, but also the properties that matter to the hosts themselves. The authors then go on to use these rejection decisions by hosts to explore error rates - accepting a foreign egg (Type II) vs. rejecting their own egg (Type I) and identify that the error rates differ among host species depending on which signature strategy has been adopted. Finally, simulations are used to identify the signal distances at which rejection errors occur, depending on whether the receiver uses a threshold or category-based decision AND what the underlying variation in the signal has evolved to be.

The manuscript is elegantly written and for the most part (see specific comments below), much of the methodological decisions and theoretical basis for the work is explained very clearly. The most significant finding is that category-based perception is most effective when trait variation can also be categorised - this raises an interesting question as to which evolves first, perceptual ability or traits. Coevolution with a mimicking antagonist complicates this, however, as there may also be selection for intermediate traits to avoid the mimic. These are questions and problems that are generalisable across coevolving systems - and which are highlighted and nicely discussed in the manuscript.

Specific comments:

abstract sentence 3: has redundancy - if categories are not discrete, would this differ conceptually from continuous variation for recognition? Suggest for sake of clarity to modify this and the following sentence to read 'Here we test how discrete categorical distributions may influence fundamental trade-offs in signal recognition, using a brood-parasite host system.'

Line 59: consider adding emphasis to the and (e.g. by italics) to make it clearer that this is the surprising aspect of the results

Lines 76 - 80: here the use of gametes as an example to explain categorically distributed traits is useful, but I wonder if this is perhaps a little too obvious (although I understand the motivation to help relate the study to a broader audience). Here it may be more useful and pertinent to the study question to consider using a well-known example from one of the contexts mentioned for trait perception - this would help the reader make the conceptual link between these main concepts.

Lines 80-82: given the space given to describe a categorical trait, it is surprising that categorical perception is not as clearly explained. Three examples are referred to, but readers would still need to go to those citations to figure out what is meant by categorical trait perception.

Lines 167-170: this wording makes it seem like the categories being consistent with our perception is a key result - is this intended? It makes it easier for future studies to refer back to perhaps, but I don't think that this deserves being repeated verbatim for each species. Perhaps it could be reworded such that the variation clusters and can therefore be described categorically (or not) (Figure 2), and then a comment that the results match human perception could be included in the otherwise very short 'paragraph' about trait correlations (lines 220-221) and avoid it becoming lost.

Figure 3: 'dumbbell plots' are not particularly informative as to the distribution of the data. Consider using e.g. violin plots for the experimental data so that the data is more clearly visualised.

Line 287: the assumption that Type I errors are only possible if a parasitic egg has been rejected is a little puzzling as there is some empirical evidence from other host species showing that they may remove their own eggs accidentally, leaving a parasitic egg behind (although it is possible that this could be an artefact of the use of artificial egg models). Could this assumption be clarified?

Simulations, text and Figure 4: I found this section a little more challenging to follow, requiring several reads to figure out what was done and why. Perhaps consider explaining explicitly in the figure legend what 'threshold distance' refers to as this is likely to trip up more naive readers, and because it also shares 'threshold' with the rejection mechanism. I got distracted by the use of the same colours across the rows of this figure, at first thinking that it didn't make sense to simulate the variation for categorical signals to have so little variation. Perhaps (d)-(f) could use different shades of red and blue to highlight that these reflect different rejection mechanisms of the respective types of signature variation.

Additional comments from the Academic Editor [lightly edited]:

I have a few very minor comments for the authors

Line 101 'depend upon'

Table 1 - add sample sizes

Line 308 change 'mass' to 'number'

Line 342. I understand the maths, but there is something strange about referring to the overall Type II error rate and including nests (the proportion being 1-p) where there was no opportunity for such an error because there were no parasitic eggs. A simple solution is to refer to the NUMBER of Type II errors (i.e. drop the 'rate'). Or just be clear that this is not the actual rate of errors when confronted by a parasitic egg.

Finally, I think the authors could protect themselves against a valid criticism by noting that this is a comparison of only a few species that focuses on egg trait distributions as the cause of differences among species in egg rejection/acceptance rates. It might be unlikely, but it is not inconceivable, that some other factor that differs among the species causes differences among these species in rejection/acceptance rates. To take a rather silly example, maybe all the species are equally good at discrimination between host and parasite eggs. But the species might vary in their propensity to act on this information because removing eggs increases the risk of total nest failure if predators detect removed eggs and use this information to locate nests. If the occurrence of such predators differed across species it could cause differences in type 1 and type 2 error rates. I am sure the authors can think of a more biological plausible alternate explanation for their findings.

---

## [Editor Report · Decision Letter 2]

6 Feb 2026

Dear Dr Dixit,

Thank you for the submission of your revised Short Report "Categorical identity signatures can reduce host error rates during brood parasitism" for publication in PLOS Biology. On behalf of my colleagues and the Academic Editor, Michael Jennions, I am pleased to say that we can in principle accept your manuscript for publication, provided you address any remaining formatting and reporting issues. These will be detailed in an email you should receive within 2-3 business days from our colleagues in the journal operations team; no action is required from you until then. Please note that we will not be able to formally accept your manuscript and schedule it for publication until you have completed any requested changes.

PRESS

Sincerely,

Taylor

Taylor Hart, PhD,

Associate Editor

PLOS Biology

thart@plos.org